# Inflammatory and Anti-Inflammatory Parameters in PCOS Patients Depending on Body Mass Index: A Case-Control Study

**DOI:** 10.3390/biomedicines11102791

**Published:** 2023-10-14

**Authors:** Elena Vasyukova, Ekaterina Zaikova, Olga Kalinina, Inga Gorelova, Irina Pyanova, Elena Bogatyreva, Elena Vasilieva, Elena Grineva, Polina Popova

**Affiliations:** 1Institute of Endocrinology, Almazov National Medical Research Centre, 194156 Saint Petersburg, Russia; 2World-Class Research Center for Personalized Medicine, Almazov National Medical Research Centre, 194156 Saint Petersburg, Russia; 3Department of Obstetrics and Gynecology, Almazov National Medical Research Centre, 194156 Saint Petersburg, Russia; 4Scandinavia AVA-PETER Clinic, 191014 Saint Petersburg, Russia; bogatyreva-elena@yandex.ru; 5Central Clinical Diagnostic Laboratory, Almazov National Medical Research Centre, 194156 Saint Petersburg, Russia; elena-almazlab@yandex.ru

**Keywords:** IL-1 α, IL-1 RA, IL-2, IL-3, IL-4, IL-5, IL-6, IL-9, IL-13, IL-15, IL-17E, IL-18, SCD40L, TNF α, TNF β, FKN fractalkine, GROa, MIG, MCP-1, MCP-3, MIP-1 α, MIP-1 β

## Abstract

Background: it has been suggested that chronic low-grade inflammation plays an important role in the pathogenesis of polycystic ovary syndrome (PCOS). According to previous studies, it remains unclear which cytokines influence the development of this syndrome and whether their increase is associated with the presence of excess weight/obesity or is an independent factor. The aim of our research was to determine the parameters of chronic inflammation in women with PCOS in comparison with healthy women in the normal weight and the overweight subgroups. Methods: This case-control study included 44 patients with PCOS (19 women with a body mass index (BMI) < 25 kg/m² and 25 women with a BMI ≥ 25 kg/m²) and 45 women without symptoms of PCOS (22 women with a BMI < 25 kg/m² and 23 women with a BMI ≥ 25 kg/m²). Thirty-two cytokines were analyzed in the plasma of the participants using Immunology multiplex assay HCYTA-60K-PX48 (Merck Life Science, LLC, Germany). Results: Cytokines: interleukin-1 receptor antagonist (IL-1 RA), IL-2, IL-6, IL-17 E, IL-17 A, IL-18, and macrophage inflammatory protein-1 alpha (MIP-1 α) were increased in women with PCOS compared to controls, both in lean and overweight/obese subgroups (*p* < 0.05). Moreover, only lean women with PCOS had higher levels of IL-1 alpha, IL-4, IL-9, IL-12, IL-13, IL-15, tumor necrosis factor (TNF- α) alpha and beta, soluble CD40 and its ligand (SCD40L), fractalkine (FKN), monocyte-chemotactic protein 3 (MCP-3), and MIP-1 β compared to the control group (*p* < 0.05). IL-22 was increased in the combined group of women with PCOS (lean and overweight/obese) compared to the control group (*p* = 0.012). Conclusion: Chronic low-grade inflammation is an independent factor affecting the occurrence of PCOS and does not depend on the presence of excess weight/obesity. For the first time, we obtained data on the increase in such inflammatory parameters as IL-9, MCP-3, and MIP-1α in women with PCOS.

## 1. Introduction

Polycystic ovary syndrome (PCOS) is a common endocrine disease in women of reproductive age. It is characterized by clinical or biochemical hyperandrogenism, oligo-/anovulation, and polycystic ovaries detected by ultrasound examination [1].

Numerous studies have been aimed at studying the cause of PCOS. The most studied pathogenetic factors include insulin resistance (IR) and compensatory hyperinsulinemia (GI). About 75% of women with PCOS have IR [2]. Insulin and luteinizing hormone (LH) act synergistically on theca cells, stimulating ovarian androgen production [3]. It is also necessary to note the special role of obesity in PCOS and IR. Obesity increases IR and GI, which in turn exacerbates PCOS symptoms, while weight loss and exercise lower IR and GI, improving symptoms of PCOS [4].

Recently, the effect of chronic low-grade inflammation on PCOS has been actively discussed. The mechanism of chronic low-grade inflammation is explained through the existing correlations between PCOS and changes in immune cells. In ovaries and adipose tissue, macrophages are the most abundant immune cells balancing destructive and protective cell-mediated immunity in inflammation [5]. Obesity and IR promote the transition of macrophages from the anti-inflammatory state M2 to the pro-inflammatory state M1, which leads to the production of interleukin-1 (IL-1), interleukin-6 (IL-6), and tumor necrosis factor-alpha (TNF- α) [6]. Obesity is associated with increased levels of lipids in the blood, including non-esterified fatty acids (NEFA). The inflammatory response may be due to lipotoxicity, which occurs when these acids accumulate in the blood [7]. Studies of the intestinal microbiome in PCOS put forward the theory that insulin resistance occurs when the ratio of Gram-positive to Gram-negative bacteria changes, in which the latter become more numerous, and the LPS released by them causes endotoxemia in the blood, which in turn activates macrophages and disrupts the function of the insulin receptor [8].

According to a recent meta-analysis, women with PCOS had elevated levels of C-reactive protein (CRP) and IL-6 compared to healthy women, while no difference in TNF- α was found [9]. Interestingly, higher levels of CRP were found in women with PCOS regardless of obesity [9].

The pro-inflammatory effect of chemokines is also being actively studied in PCOS. The most studied chemokines in PCOS are monocyte chemoattractant protein 1 (MCP-1) and fractalkine (FKN) [10]. A recent meta-analysis revealed an increase in MCP-1 in both obese and lean women with PCOS [11].

In addition to cytokines with pro-inflammatory effects, there are cytokines with anti-inflammatory effects. The most studied of them in PCOS are interleukin-1 receptor antagonist (IL-1RA), interleukin 10 (IL-10), and interleukin-4 (IL-4) [12,13,14,15,16,17,18,19,20].

The anti-inflammatory IL-1RA is secreted by various cell types, binds to the IL-1 receptor, and thus blocks pro-inflammatory IL-1 signaling by both interleukin-1 alpha (IL-1α) and interleukin-1 beta (IL-1β) [12]. The possibility of blocking IL RA ovulation has been demonstrated in in vitro and in vivo studies. A study in mice demonstrated the effect of IL-1RA on ovulation inhibition [13]. In vitro, IL-1RA weakened the effect of human chorionic gonadotropin and follicle-stimulating hormone (FSH) on the expansion of cumulus and the synthesis of follicular hyaluronic acid [14]. Luotola K. et al. (2016) showed that IL-1RA was elevated in PCOS compared with controls but the difference disappeared after adjusting for body mass index (BMI) [15].

IL-10 is a cytokine produced by Th2 cells, T cells, and monocytes/macrophages [16]. IL-10 inhibits the action of such pro-inflammatory cytokines as IL-6, IL-1, and TNF- α [17]. Several studies have reported a relationship between IL-10 polymorphism and PCOS [16,18].

It is possible that an increase in anti-inflammatory cytokines in PCOS is a compensatory response to an increase in pro-inflammatory cytokines.

Apart from the parameters of inflammation with studied effects on PCOS, there are parameters of inflammation whose role in PCOS remains unclear, however, their impact on IR or obesity is known. These cytokines include interleukin-9 (IL-9), interleukin-3 (IL-3), monokine induced by gamma interferon (MIG), chemokine ligand 1 alpha (GRO a), macrophage inflammatory protein-1 alpha (MIP-1a) [21,22,23,24].

We found only one study that showed a reduced level of IL-7 in women with PCOS compared to the control group, but the groups were not comparable (the group with PCOS included 21 patients, while the control group included 120 patients) [25].

There are also contradictory data on the effects of IL-22 in PCOS. The study by Aksun S. et al. (2023) did not show a significant difference in the level of IL-22 between PCOS and control [26], whereas Zhou H. et al. (2022) revealed that IL-22 levels were lower in women with PCOS [27].

Another limitation of the studies on the relationship of PCOS and chronic low-grade inflammation is the lack of correction for the presence of excess weight/obesity [14,26,27].

The purpose of our study was to investigate parameters of chronic low-grade inflammation in women with PCOS compared with healthy women in the overweight and normal weight subgroups.

## 2. Materials and Methods

The study was conducted at the Almazov National Medical Research Center as part of the “Development of a system for personalized information support for patients with polycystic ovary syndrome” project, which started in January 2021.

### 2.1. Design and Study Population

The study included 44 patients with PCOS (19 women with a body mass index (BMI) < 25 kg/m² and 25 women with a BMI ≥ 25 kg/m²) and 45 normally ovulating women without PCOS (22 women with a BMI < 25 kg/m² and 23 women with BMI ≥ 25 kg/m²) (control group) matched by age.

The Rotterdam criteria were used to diagnose PCOS [1]. The study included women from 18 to 40 years old. The control group consisted of women with a regular menstrual cycle (more than 21 days and less than 35 days), absence of hyperandrogenism, and normal ovarian morphology by ultrasound examination.

Patients were not included in the study in the presence of other causes of menstrual cycle disorders, such as congenital hyperplasia of the adrenal cortex, hyperprolactinemia, hypothyroidism or thyrotoxicosis, as well as pregnancy and breastfeeding, diabetes mellitus, and regular intake of drugs affecting the function of the ovaries, adrenal glands, and carbohydrate metabolism for 2 months prior the study.

### 2.2. Hormonal Evaluation

Blood sampling for hormonal evaluation was performed in the morning on days 3–5 of menstrual cycle or a progestin-induced withdrawal bleeding and serum was frozen at −80 °C. Enzyme immunoassay was used to determine the levels of insulin, sex steroid binding globulin (SHBG), FSH, and LH (Elecsys 2010, Roche-Diagnostics, Mannheim, Germany). Reference values for this method are 1–10 mIU/mL for LH, 1.8–10.5 mIU/mL for FSH (in follicular phase), 0.29–1.67 nmol/L for testosterone, 26.10–110.00 nmol/L for SHBG, and 2.6–24.9 µU/mL for insulin. The level of serum androstenedione was determined by the enzyme immunoassay method using reagents from DRG, Germany, with reference values of 0.75–3.89 ng/mL and 0.1–0.8 ng/mL (for women in the follicular phase). Determination of serum leptin (EIA, London, Canada) was performed by manual enzyme immunoassay. Reference values according to this method are 3.63–11.09 ng/mL.

### 2.3. Biochemical Evaluation

The metabolic profile included fasting plasma glucose (FPG), lipidogram, insulin, and leptin. Serum lipid levels were determined on a biochemical analyzer COBAS INTEGRA 400 (Roche, Basel, Switzerland), using reagents, calibrators, and controls from the same company. Reference values for this method are 0.0–5.17 mmol/L for total cholesterol, 0.0–1.69 mmol/L for triglycerides, 1.04–1.55 mmol/L for high-density lipoprotein cholesterol (HDL-C), 0.10–1.0 mmol/L, 2.69–4.11 mmol/L for low-density lipoprotein (LDL) cholesterol. IR was determined by the model HOMA-IR using the following formula: (FPG, mmol/L × fasting insulin, μIU/mL)/22.5 [28].

### 2.4. Cytokines and Inflammatory Parameters Analysis

Thirty-two cytokines and inflammatory parameters (IL-1, IL-1a, IL-1b, IL-1 RA, IL-2 (interleukin 2), IL-3, IL-4, IL-6, IL-9, IL-10, IL-13 (interleukin 13), IL-15 (interleukin 15), IL-17 A (interleukin 17 alpha), IL-17 E (interleukin 17 E), IL-17 F (interleukin 17 F), IL-18 (interleukin 18), SCD40L (soluble CD40) and its ligand, TNF- α, TNF-β (tumor necrosis factor beta), FKN, GROa, MIG, MCP-1, MCP-3 monocyte-chemotactic protein 3, MIP-1a, and MIP-1b (macrophage inflammatory protein-1 beta)) were assessed by multiplex analysis using the MILLIPLEX^®^ MAP Human Cytokine/Chemokine/Growth Factor Panel A (HCYTA-60K-PX48, Merck Life Science, LLC, Darmstadt, Germany) according to the manufacturer’s instructions. Overnight incubation at 4 °C and a handheld magnet were utilized. The plates were analyzed on a Luminex MAGPIX System (Luminex Corporation, Austin, TX, USA) and the data were generated with xPONENT software 4.3. Luminex^®^ uses proprietary techniques to internally color-code microspheres with two fluorescent dyes. Using the concentration of these dyes, sets of beads coated with capture antibody were created. After capturing the analyzed sample by the bead, a biotinylated detection antibody was injected. The reaction mixture was then incubated with the streptavidin-PE conjugate to complete the reaction on the surface of each microsphere. Each individual microsphere was identified and the result of its bioassay is quantified based on fluorescent reporter signals. The results were presented in picograms per milliliter (pg/mL).

### 2.5. Statistical Analysis

Statistical analysis was carried out using SPSS 26.0 software (SPSS Inc., Chicago, IL, USA). The distribution was different from normal, so the data of clinical (age, BMI), biochemical (FPG, total cholesterol), hormonal parameters (insulin, sex hormones), and cytokines are presented in the form of medians and interquartile ranges. The Mann–Whitney test was used for their intergroup comparison. Correlation analysis of plasma cytokine levels was performed using Spearman correlation coefficient.

## 3. Results

### 3.1. Clinical, Biochemical, and Hormonal Parameters

Table 1 presents data on clinical, biochemical, and hormonal parameters. Women with PCOS did not differ from the control group in terms of age and BMI. PCOS group had a considerably longer duration of menstrual cycle (MC).

The levels of total cholesterol, insulin, LH, total testosterone, androstenedione, and the free androgen index were higher, while SHBG and FSH were lower in women with PCOS. Interestingly, FPG levels were lower in PCOS group compared to healthy women (medians were 4.8 vs. 5.0 mmol/L, *p* = 0.020), which may be explained by the fact that overweight/obese women with PCOS followed a diet before the study.

In the subgroup analysis, only lean women with PCOS had higher levels of cholesterol and its fractions (LDL and HDL) compared to controls, which was not the case in the overweight/obese subgroup. On the contrary, insulin and HOMA-IR index were higher only in the overweight women with PCOS. The levels of total testosterone, FAI and androstenedione were increased in women with PCOS regardless of BMI.

### 3.2. Comparison of the Interleukin Levels

The interleukin levels of all the participants are summarized in Table 2. Cytokines, namely IL-1 RA, IL-2, IL-6, IL-17 E, IL-17 A, IL-18, and MIP-1a were higher in women with PCOS, regardless of whether they were overweight/obese (*p* < 0.05). In the subgroup analysis, only lean women with PCOS had higher levels of IL-1a, IL-3, IL-4, IL-8, IL-9, IL-12, IL-13, IL-15, TNF- α and -β, SCD40L, FKN, Eotaxin, MCP-3, and MIP-1b compared to the control group (*p* < 0.005). IL-22 and IL-17-F were elevated in PCOS, but this difference was not preserved in the subgroups. The levels of IL-10, IL-7, IL-1 b, IL-5, IL-27, and GRO a did not differ between the groups.

### 3.3. Chemokines and Other Inflammatory Parameters

The levels of SCD40L, TNF- α, TNF-β, FKN, MCP-3, and MIP-1β were increased in women with PCOS due to the difference in the lean subgroup. The level of MIP-1-α was higher both in the lean and overweight patients with PCOS (Table 3).

### 3.4. Associations of Inflammatory Parameters and Metabolic Parameters

Direct correlations have been found between IL-1 RA and insulin (r = 0.335, *p* = 0.024), HOMA-IR index (r = 0.311, *p* = 0.037), weight (r = 0.483, *p* = 0.001) and between IL 6 and insulin (r = 0.550, *p* < 0.001), HOMA-IR index (r = 0.532, *p* < 0.001), weight (r = 0.541, *p* < 0.001), fasting blood glucose (r = 0.327, *p* = 0.028).

### 3.5. IR and Inflammation in PCOS

To study the effect of IR on inflammation in PCOS, we divided women with PCOS according to the HOMA-IR index. A value of more than 2.5 confirmed the presence of IR. Of the 44 women with PCOS, IR was detected in 20 participants. The levels of IL-1 RA and IL-6 were higher in the subgroup with IR (*p* < 0.001) (Table 4).

Among chemokines, a difference was detected only for Eotaxin, which was significantly lower (median 37.4 versus 53.9) in the subgroup with IR (Table 5).

## 4. Discussion

Our study provides evidence of low-grade chronic inflammation in women with PCOS independent of obesity as supported by an increase in 24 inflammatory parameters, 3 of which were demonstrated for the first time to be associated with PCOS: IL-9, MCP-3, and MIP-1α. We also confirmed the connection of IL-22 with PCOS; the data on which were previously contradictory.

IL-9, a pleiotropic cytokine with diverse functions, was first identified as a T-cell and mast-cell growth factor [29]. It has been shown to exert both pro-inflammatory- and anti-inflammatory-cell-type specific effects and is implicated in the pathogenesis of asthma, inflammatory bowel diseases, autoimmune encephalomyelitis, and other allergic and autoimmune diseases. On the other hand, IL-9 has beneficial effects in initiating immunity against helminth infection and tumors [30]. It has not been studied in the conditions of insulin resistance, type 2 diabetes mellitus, or obesity and its role in the pathogenesis of PCOS requires further investigation.

Chemokines MCP-3 and MIP 1-α contribute to the development of obesity, but their exact role has not been determined [31,32]. In our study, lean women with PCOS had higher levels of MIP 1-α and MCP-3 compared to the control group.

Experiments on obese mice demonstrated restoration of insulin secretion and improvement of insulin sensitivity after administration of IL-22 [33]. The result could be useful for the treatment of PCOS in humans. According to previous studies, IL-22 did not differ in women with PCOS and was even lower, which does not agree with the data of our study [26,27].

Our other goal was to clarify the association of weight with the increase in cytokines in PCOS. According to our data, the association of IL-1α, IL-3, IL-4, IL-8, IL-9, IL-12, IL-13, IL-15, TNF-α and -β, SCD40L, FKN, Eotaxin, MCP-3, and MIP-1β with PCOS was observed due to a subgroup with normal weight, while IL-1 RA, IL-2, IL-6, IL-17 E, IL-17 A, IL-18, and MIP-1α were elevated in PCOS regardless of BMI.

The exact mechanism of inflammation in PCOS is currently unclear. The role of obesity as a pro-inflammatory factor is the most studied one as it is often found in women with PCOS. Obesity-induced chronic low-grade inflammation is explained by hypertrophy and hyperplasia of adipocytes leading to free fatty acid release, which increases the level of pro-inflammatory cytokines in macrophages through Toll-like receptors [34].

Inflammation in lean women with PCOS could have other pathogenic mechanisms, but most studies of parameters of chronic low-grade inflammation have been conducted in overweight or obese patients with PCOS. A study of healthy, lean women without PCOS treated with dehydroepiandrosterone sulfate (DHEA-S) showed an increase in inflammatory parameters after an increase in androgen levels [35]. Thus, it can be suggested that the cause of chronic low-grade inflammation in lean patients is hyperandrogenemia.

It is also hypothesized that the chronic low-grade inflammation in PCOS may have a genetic basis. The hypothesis is based on the results of few studies reporting an association between PCOS and pro-inflammatory genotypes encoding IL-6, type 2 TNF receptor, and TNF-α [36,37,38].

Our data on the increase in pro-inflammatory parameters in PCOS are consistent with previous studies of such parameters as IL-1α, IL-18, and IL-8 [39,40,41].

We found only one study of serum IL-2 levels in patients with PCOS, in which IL-2 levels were higher in obese women with PCOS than in lean women with PCOS, while no differences were found between PCOS and control groups. However, the limitation of the study is a small sample of patients (19 women with PCOS and 20 controls) [42]. In our study, IL-2 levels were higher in PCOS, regardless of BMI.

A meta-analysis conducted by Peng et al. (2016) showed that elevated IL-6 levels in women with PCOS were associated with IR and androgen levels. In addition, elevated IL-6 levels were found in both lean and obese patients with PCOS [43]. Our data are consistent with data from previous studies. We also found direct correlations between IL-6 and insulin, HOMA-IR index, weight, and fasting blood glucose levels.

We found no statistically significant difference in the levels of IL-5, IL-7, IL-27, and IL-10 between PCOS and the control group.

Previously, a study by Benson et al. (2008) showed a significant decrease in IL-5 levels in patients with PCOS, regardless of BMI [19].

Nehir et al. (2016) showed that IL-27 levels were significantly lower in lean patients with PCOS compared to the control group [44].

In a study by Talaat et al. (2016), IL-10 levels were lower in women with PCOS [18].

The previous studies were limited by a small sample size.

In our study, 48% of the overweight/obese women with PCOS reported following a diet which could affect the levels of cytokines in this subgroup. Mehrabani et al. (2012) showed a decrease in the level of inflammation in women following a low-calorie diet with a high protein content and foods with a low glycemic index [45].

We did not analyze the more commonly used inflammatory markers, such as CRP or procalcitonin. However, we provide indirect evidence of increased CRP through the increased levels of IL-6 and TNF-α, which are the main mediators of CRP production in the liver. According to two large meta-analyses by Escobar-Morreale et al. (2010) and Aboeldalyl et al. (2021), women with PCOS have elevated CRP levels regardless of obesity, which is consistent with our data [9,46]. Procalcitonin is a well-studied marker of bacterial infection [47]. Rashad et al. (2013) showed an increase in procalcitonin levels in women with PCOS that was dependent on the presence of overweight/obesity [48]. Thus, the results of our study and other research provide further evidence supporting the hypothesis that chronic inflammation is related to PCOS pathology independent of obesity.

It has also been established that insulin sensitivity is reduced by ~40% in PCOS women, independent of obesity [49], although obesity further impairs insulin metabolism. TNF-α and IL-6, two main adipose-tissue-derived inflammatory cytokines, have been implicated in the induction of insulin resistance [50,51].

In our study, the relation between IR and PCOS was detected only for IL-6, IL-1 RA and Eotaxin. IL-6 induces insulin resistance by downregulating insulin receptor substrate-1 (IRS-1) phosphorylation and transcription [52,53] and by upregulating suppressor of cytokine signaling 3 (SOCS3), which inhibits insulin receptors [54]. The involvement of IL-1 in regulation of insulin sensitivity is highlighted by a recent observation that the IL-1 blocking agent improved IR in rheumatoid arthritis patients with comorbid type 2 diabetes (T2D) [55].

A hypothesis explaining a causal relation between chronic inflammation and insulin resistance in PCOS was suggested by Shorakae et al. (2018), who investigated the links between insulin resistance, hyperandrogenism, and chronic low-grade inflammation in PCOS in a cross-sectional study. The authors concluded that PCOS-related hyperandrogenism leads to adipose tissue inflammation with subsequent dysregulated adipokine secretion resulting in insulin resistance [56].

A plausible reason for the lack of association between IR and the majority of cytokines in our study is that the Homa-IR index is not sensitive enough to detect IR, especially in lean people with PCOS. It is also possible that the effects of other inflammatory parameters altered in PCOS are mediated through other mechanisms unrelated to IR.

The limitation of our study is the observational nature and a cross-sectional design. Experimental studies are required to clarify the pathways underlying the effects of multiple inflammatory cytokines in PCOS.

## 5. Conclusions

In recent years, chronic low-grade inflammation, despite its unclear origin, has become widely discussed as an important factor in the pathogenesis of PCOS. In our study, for the first time, an increase in three parameters of chronic low-grade inflammation was detected: IL-9, MCP-3, and MIP-1α. We also confirmed the association with PCOS for another 21 inflammatory parameters previously studied in the condition. Through examining inflammatory parameters separately in overweight and lean women with PCOS we provide evidence that an increase in inflammatory parameters is detected regardless of the obesity.

## Figures and Tables

**Table 1 biomedicines-11-02791-t001:** Clinical and laboratory characteristics of the subjects.

	BMI < 25				*p*	BMI ≥ 25		*p*	PCOS (*n* = 44)	Control (*n* = 45)	*p*
	PCOS (*n* = 19)		Controls (*n* = 22)			PCOS (*n* = 25)	Controls (*n* = 23)				
Age, years	27 (24–30)		27 (25–29)		0.981	27 (22.5–30.5)	30 (28–30.5)	0.062	28 (24–31)	26 (26–30)	0.898
BMI	21.9 (20–23.8)		21.5 (20–22.5)		0.231	33.9 (28.2–37.9)	28.7 (27.3–33.1)	0.065	25.3 (21.7–32.5)	26 (21.6–28.7)	0.407
Average duration of MC, days	30 (28–37)		28 (28–29)		0.007	30 (28–50)	28 (28–29)	0.007	55 (32–90)	28 (28–29)	<0.001
Dieting before inclusion in the study	0%	0%	48%	0%	0.001						
FPG, mmol/L	4.7 (4.5–5.1)		4.9 (4.6–5.1)		0.690	5.0 (4.8–5.2)	5.3 (5.0–5.5)	0.037	4.8 (4.5–5.2)	5.0 (4.8–5.3)	0.020
Insulin, μIU/mL	7.0 (4.3–8.7)		6.4 (4.8–8.8)		0.888	18.2 (12.4–23.7)	9.5 (6.7–17.0)	0.001	9.4 (6.4–18.6)	8.0 (5.9–10.6)	0.042
Leptin, ng/mL	18.4 (11.3–35.6)		16.3 (9.3–48.7)		0.606	40.2 (30–89.7)	37.6 (25.2–48.8)	0.318	31.5 (13.3–47)	26 (9.4–44.7)	0.341
HOMA-IR index	1.4 (1–1.8)		1.4 (0.9–2)		0.925	4.3 (2.7–5.4)	2.1 (1.5–3.8)	0.003	2.1 (1.4–4.3)	1.8 (1.3–2.3)	0.117
Total cholesterol, mg/d	4.9 (4.3–5.4)		4.2 (0.6–4.9)		0.003	4.4 (3.8–5.3)	4.7 (4.3–5.1)	0.606	4.8 (4.2–5.3)	4.4 (3.7–5)	0.009
LDL-C, mg/dL	2.9 (2.7–3.4)		2.5 (1.7–2.9)		0.008	2.9 (2.1–3.6)	2.8 (2.5–3.1)	0.818	2.9 (2.2–3.3)	2.6 (2.2–3)	0.099
HDL-C, mg/dL	1.5 (1.3–1.8)		1.3 (0.3–1.6)		0.046	1.1 (1–1.4)	1.4 (1.1–1.5)	0.114	1.4 (1.1–1.7)	1.4 (1.1–1.6)	0.365
Triglycerides, mg/dL	0.62 (0.6–0.8)		0.70 (0.4–1.3)		0.324	1.0 (0.9–1.4)	0.86 (0.6–1.2)	0.065	0.9 (0.6–1.2)	0.7 (0.6–1.1)	0.146
FSH, mIU/mL	6.0 (5.0–6.9)		6.7 (5.8–8.1)		0.058	4.8 (4.4–5.7)	6.5 (5.6–7.7)	<0.001	6.3 (5.2–7.1)	6.7 (5.8–8.4)	0.020
LH, mIU/mL	11.1 (6.7–14.6)		5.9 (4.8–7.3)		<0.001	7.4 (6.6–12.1)	6.3 (4.6–7.6)	0.050	9.6 (6.5–12.9)	6.2 (4.7–7.8)	<0.001
Total testosterone, nmol/L	1.5 (1.2–2.1)		0.8 (0.7–1.2)		<0.001	1.8 (1.4–2.3)	1.1 (0.7–1.3)	<0.001	1.6 (1.1–2.1)	0.9 (0.7–1.2)	<0.001
Androstenedion, ng/mL	3.0 (2.3–4.9)		2.1 (1.3–2.7)		0.005	3.5 (2.2–4.6)	2.2 (1.8–3.4)	0.018	3.4 (2.7–5.0)	2.2 (1.7–2.9)	<0.001
SHBG, nmol/L	57.0 (40.6–80.1)		82.4 (55.9–102.4)		0.062	26.7 (15.9–41.8)	59.0 (40.3–78.4)	0.001	46.9 (29.7–75)	67.5 (51.4–95.4)	<0.001
FAI, %	2.8 (1.8–3.8)		1.0 (0.9–1.4)		0.003	5.0 (3.4–9.5)	1.7 (1.4–2.9)	<0.001	3.3 (1.9–5.5)	1.4 (0.9–2.1)	<0.001

BMI—Body mass index, FAI—free androgen index, FBG—fasting blood glucose, FSH—follicle-stimulating hormone, HDL-C—high-density lipoprotein cholesterol, HOMA-IR index—homeostasis model assessment of insulin resistance, LDL-C—low-density lipoprotein cholesterol, LH—luteinizing hormone, MC—menstrual cycle, PCOS—polycystic ovary syndrome, SHBG—sex-hormone-binding globulin.

**Table 2 biomedicines-11-02791-t002:** Comparison of the interleukin levels of the subjects.

Interleukins	BMI < 25		*p*	BMI ≥ 25		*p*	PCOS (*n* = 44)	Control (*n* = 45)	*p* **	*p* ***
	PCOS (*n* = 19)	Controls (*n* = 22)		PCOS (*n* = 25)	Controls (*n* = 23)					
IL-1a	1.3 (0.5–6.9)	0 (0–3.2)	0.006	2.56 (0.0–5.92)	0.51 (0.0–5.92)	0.358	1.3 (0.5–5.9)	0 (0–2.5)	0.014	0.723
IL-1b	11.7 (5.8–21.7)	5.2 (0.7–16.9)	0.124	13.27 (3.62–21.54)	5.86 (1.74–17.41)	0.230	13.2 (5.2–21.7)	5.8 (1.7–16.9)	0.052	0.776
IL-1 RA	1.0 (0.9–1.9)	0.5 (0.3–0.8)	0.001	1.95 (1.34–2.75)	0.96 (0.69–1.12)	<0.001	1.5 (0.9–2.2)	0.8 (0.4–1.0)	<0.001	0.004
IL-2	0.28 (0–1.5)	0 (0–0.6)	0.004	0.43 (0.0–0.85)	0.0 (0.0–0.14)	0.012	0.43 (0–1.14)	0 (0–0.11)	<0.001	0.711
IL-3	0.7 (0–0.2)	0.0 (0.0–0.0)	0.030	0.09 (0.0–0.29)	0.05 (0.0–0.22)	0.484	0.05 (0–0.6)	0 (0–0.09)	0.053	0.814
IL-4	0.5 (0–0.6)	0.0 (0.0–0.0)	0.008	0.05 (0.0–0.54)	0.0 (0.0–0.0)	0.094	0.05 (0–0.6)	0 (0.0–0.0)	0.002	0.719
IL-5	1.99 (1.24–3.35)	1.29 (0.84–3.1)	0.184	1.94 (1.39–3.99)	1.59 (0.94–4.05)	0.448	1.9 (1.3–3.6)	1.4 (0.9–3.5)	0.144	0.937
IL-6	0.48 (0.24–1.12)	0.0 (0.0–0.24)	0.004	1.2 (0.58–2.17)	0.32 (0.0–0.64)	0.002	0.7 (0.4–1.2)	0.07 (0–0.48)	<0.001	0.009
IL-7	0.75 (0.08–1.05)	0.08 (0.0–0.75)	0.050	0.33 (0.07–0.81)	0.27 (0.21–1.19)	0.644	0.4 (0.08–0.99)	0.2 (0–0.87)	0.268	0.345
IL-8	1.56 (1.12–2.63)	0.98 (0.66–1.56)	0.034	1.43 (0.95–1.83)	1.77 (1.08–2.23)	0.116	1.5 (1.0–2.1)	1.5 (0.8–1.97)	0.534	0.146
IL-9	6.71 (0.0–12.01)	0.0 (0.0–3.99)	0.048	5.26 (0.0–11.35)	0.0 (0.0–2.56)	0.060	5.7 (0–11.4)	0 (0–3.9)	0.007	0.800
IL-10	0.0 (0.0–0.0)	0.0 (0.0–0.0)	0.555	0.0 (0.0–0.0)	0.0 (0.0–0.0)	0.669	0.0 (0.0–0.0)	0.0 (0.0–0.0)	0.967	0.963
IL-12	22.53 (13.62–31.02)	14.84 (7.27–19.62)	0.012	21.95 (17.25–25.4)	18.44 (9.87–27.09)	0.176	21.9 (16–28.7)	16 (9.8–20.7)	0.005	0.883
IL-13	75.93 (54.9–194.96)	19.64 (1.22–54.9)	<0.001	58.98 (31.58–128.68)	46.26 (7.6–106.89)	0.373	74.1 (43.9–143.5)	36.7 (3–81.2)	0.001	0.117
IL-15	7.8 (4.98–8.61)	4.8 (3.23–6.32)	0.006	7.64 (6.49–9.96)	7.31 (5.32–8.12)	0.235	7.6 (5.6–9.5)	6.3 (4.2–7.6)	0.004	0.509
IL-17A	5.33 (1.57–8.82)	0.57 (0.29–4.43)	0.002	5.62 (1.98–8.82)	1.57 (0.29–5.03)	0.033	5.6 (1.5–8.8)	0.5 (0.2–4.4)	<0.001	0.882
IL-17E/IL-25	310.25 (172.63–529.23)	154.83 (52.69–205.93)	0.007	352.69 (221.73–676.94)	172.63 (117.52–366.08)	0.013	352.6 (205.9–643.9)	172.6 (52.6–325)	<0.001	0.547
IL-17F	3.43 (1.13–7.21)	0.73 (0.0–3.65)	0.066	3.0 (1.33–7.21)	1.13 (0.35–4.54)	0.083	3.2 (1.1–7.2)	0.7 (0–3.6)	0.011	0.856
IL-18	14.93 (9.82–20.5)	8.84 (3.51–13.92)	0.025	16.29 (10.37–24.37)	10.18 (5.78–17.41)	0.049	14.9 (10.1–22.9)	9.7 (4.5–16.4)	0.003	0.570
IL-22	0.0 (0.0–121.66)	0.0 (0.0–0.0)	0.060	0.0 (0.0–26.1)	0.0 (0.0–0.0)	0.097	0.0 (0.0–29.1)	0.0 (0.0–0.0)	0.012	0.458
IL-27	1478.66 (1190.23–1763.21)	1320.66 (1070.85–1529.68)	0.205	1073.96 (772.69–1476.52)	1303.77 (1022.43–1803.25)	0.124	1266 (967.7–1703.3)	1303.7 (1058–1652.4)	0.768	0.051

IL—interleukin, IL-1a—interleukin alpha, IL-1b—interleukin beta, IL-1 RA—interleukin-1 receptor antagonist. Note: The interleukin levels are presented in picograms per milliliter (pg/mL); **—comparing PCOS versus control without division by BMI; ***—comparing women with PCOS with BMI < 25 kg/m² and BMI ≥ 25 kg/m².

**Table 3 biomedicines-11-02791-t003:** Comparison of the tumor necrosis factors, SCD40L, and chemokines of the subjects.

Analytes	BMI < 25		*p*	BMI ≥ 25		*p*	PCOS (*n* = 44)	Control (*n* = 45)	*p* **	*p* ***
	PCOS (*n* = 19)	Controls (*n* = 22)		PCOS (*n* = 25)	Controls (*n* = 23)					
SCD40L	246.1 (158.6–451.6)	124.4 (91.5–206)	0.009	252.66 (154.75–400.17)	146.92 (113.75–364.25)	0.170	252.6 (158.6–428)	134.8 (109.3–337.5)	0.004	0.901
TNF-a	16.5 (12.1–45)	7.2 (5.2–11.5)	<0.001	12.73 (10.32–18.59)	9.71 (8.62–13.62)	0.066	15.6 (11.5–21.4)	9.4 (5.9–12.1)	<0.001	0.020
TNF-b	8.8 (5.2–24)	2.5 (0.6–7.2)	0.004	6.11 (2.23–12.11)	3.04 (1.72–9.59)	0.230	8.4 (2.7–16.1)	3 (0.6–7.8)	0.003	0.173
FKN	153.61 (130.79–266.05)	89.67 (72.1–120.33)	<0.001	134.18 (105.58–168.95)	113.08 (81.14–150.5)	0.202	140.8 (113–216.4)	97.8 (81.1–134.1)	<0.001	0.061
MIG	656.81 (497.31–825.76)	470.75 (427.05–740.17)	0.067	689.96 (590.98–784.09)	1047.81 (687.13–1238.94)	0.025	687 (562–808)	695.5 (433–1152)	0.878	0.910
GRO-α	1.27 (0.28–3.05)	0.51 (0.0–1.95)	0.166	0.6 (0.24–1.59)	0.42 (0.0–3.22)	0.973	0.8 (0.2–1.8)	0.5 (0–2.8)	0.381	0.189
MCP-1	180.19 (141.11–192.8)	132.52 (112.04–158.57)	0.006	153.7 (106.37–183.32)	168.61 (153.07–199.03)	0.059	161 (123.8–191)	153 (121.4–181.7)	0.732	0.109
MCP-3	9.64 (6.83–37.35)	2.77 (0.0–7.38)	<0.001	8.44 (4.95–18.02)	6.26 (1.03–14.18)	0.165	8.4 (6.2–21.8)	5 (0.6–8.4)	<0.001	0.328
MIP-1 α	15.93 (8.59–29.14)	4.3 (0.77–14.07)	0.018	16.98 (10.64–25.47)	6.12 (0.77–16.2)	0.026	16.2 (8.5–26.2)	6 (0.7–14)	0.001	0.919
MIP-1 β	17.51 (14.06–22.05)	10.98 (8.84–14.37)	0.003	16.63 (14.97–19.88)	13.07 (11.6–17.84)	0.097	17.1 (14.3–20.5)	12.3 (10.5–16.6)	0.001	0.946
Eotaxin	54.17 (45.8–64.28)	44.05 (31.55–48.9)	0.010	37.71 (30.14–44.33)	43.28 (30.03–63.13)	0.097	45.3 (37.7–61.6)	43.9 (30.8–60.7)	0.654	<0.001

SCD40L—soluble CD40 and its ligand, TNF-a—tumor Necrosis Factor alpha, TNF-b—tumor necrosis factor beta, FKN—Fractalkine, GROa—chemokine ligand 1, MIG—monokine induced gamma interferon, MCP-1—monocyte chemoattractant protein 1, MCP-3—monocyte-chemotactic protein 3, MIP-1 α—macrophage inflammatory protein 1-alpha, MIP-1 β—macrophage inflammatory protein 1-beta. Note: The analyte levels are presented in picograms per milliliter (pg/mL); **—comparing PCOS versus control without division by BMI. ***—comparing women with PCOS with BMI < 25 kg/m² and BMI ≥ 25 kg/m².

**Table 4 biomedicines-11-02791-t004:** Comparison of the interleukin levels according to the HOMA-IR index.

Interleukins	HOMA < 2.5 (*n* = 24)	HOMA ≥ 2.5 (*n* = 20)	*p*
IL-1a	1.3 (0.5–7.5)	2.5 (0–5.3)	0.593
IL-1b	11.7 (5.5–22.3)	13.2 (3.6–20.4)	0.706
IL-1 RA	1.2 (0.9–2.0)	1.8 (1.3–2.9)	0.024
IL-2	0.36 (0–0.9)	0.55 (0–1.7)	0.755
IL-3	0.09 (0–0.25)	0.05 (0–0.36)	0.893
IL-4	0.05 (0–0.76)	0 (0–0.34)	0.375
IL-5	1.99 (1.34–3.75)	1.89 (1.0–3.7)	0.663
IL-6	0.48 (0.28–1.0)	1.2 (0.6–2.7)	0.004
IL-7	0.63 (0.17–1.0)	0.39 (0.01–0.81)	0.636
IL-8	1.39 (1.0–2.5)	1.56 (0.98–2.0)	0.604
IL-9	4.3 (0–13)	5.4 (0.0–9.7)	0.649
IL-10	0.0 (0.0–0.0)	0.0 (0.0–0.0)	0.706
IL-12	20.78 (16.65–27)	22.5 (15.7–33.5)	0.494
IL-13	89 (50.5–142.2)	56.9 (31.5–143.5)	0.340
IL-15	7.9 (5.8–9.7)	7.6 (5.9–9.3)	0.841
IL-17A	5.6 (1.57–10.39)	5.6 (2.1–6.8)	0.456
IL-17E/IL-25	296.5 (172.6–552.6)	391.9 (221.7–698.7)	0.243
IL-17F	3.43 (1.13–6.76)	3.0 (1.5–7.6)	0.777
IL-18	15.1 (11.03–20.21)	15.5 (9.5–28.3)	0.604
IL-22	0.0 (0.0–43.51)	0.0 (0.0–45.30)	0.956
IL-27	1478.65 (1049.6–1733.2)	1136.88 (922.5–1598.8)	0.423

IL—interleukin, IL-1a—interleukin alpha, IL-1b—interleukin beta, IL-1 RA—interleukin-1 receptor antagonist. Note: The interleukin levels are presented in picograms per milliliter (pg/mL).

**Table 5 biomedicines-11-02791-t005:** Comparison of the tumor necrosis factors, SCD40L, and chemokines levels according to the HOMA-IR index.

Analytes	HOMA < 2.5 (*n* = 24)	HOMA ≥ 2.5 (*n* = 20)	*p*
SCD40L	193.5 (156.6–439.8)	271.6 (170.4–400)	0.494
TNF-α	16.5 (12.4–22.5)	11.5 (10–20.3)	0.069
TNF-β	9.4 (5.2–16.3)	4.9 (1.8–16.9)	0.268
FKN	147.3 (130.7–241.2)	130.7 (101.6–204.8)	0.140
MIG	685.7 (535.7–842.3)	693.7 (574.5–772.1)	0.786
GRO-α	1.14 (0.42–2.94)	0.42 (0.12–1.81)	0.126
MCP-1	166.2 (139.5–189.3)	154 (101.5–191)	0.258
MCP-3	11.9 (6.5–20.8)	8.4 (4.9–20.4)	0.457
MIP-1α	15.9 (8.0–29.6)	18.2 (10.6–25.9)	0.972
MIP-1β	16.7 (12.5–20.4)	18.6 (14.9–22.5)	0.216
Eotaxin	53.9 (44.9–63.9)	37.4 (30–45)	0.001

SCD40L—soluble CD40 and its ligand, TNF-α—tumor necrosis factor alpha, TNF-β—tumor necrosis factor beta, FKN—fractalkine, GROa—chemokine ligand 1, MIG—monokine-induced gamma interferon, MCP-1—monocyte-chemoattractant protein 1, MCP-3—monocyte-chemotactic protein 3, MIP-1α—macrophage inflammatory protein 1-alpha, MIP-1β—macrophage inflammatory protein 1-beta.

## Data Availability

The data presented in this study are available upon request from the corresponding author.

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
