# Peer review of "Inflammatory and Anti-Inflammatory Parameters in PCOS Patients Depending on Body Mass Index: A Case-Control Study"

_biomedicines, 2023, doi:10.3390/biomedicines11102791_

Round 1

Reviewer 1 Report

The manuscript investigates different inflammatory and anti-inflammatory parameters in the group of women with and without PCOS divided according to body mass index.

In my opinion the investigated inflammatory and anti-inflammatory cytokines should be treated as parameters not markers.

The results should be compared with commonly used inflammatory markers for general diagnosis, such as CRP or procalcitonin to confirm their usefulness for the determination of inflammatory state.

Furthermore, the chronic inflammatory during PCOS should be treated as chronic low-grade inflammation.  

In title anti- not anty – should be written.

verse 16 - please introduce the abbreviation of PCOS

verse 27 – delete space between p and < (p <0.05)

I can not find in paper [2] information about “About 70% of women with PCOS have IR [2]”. Additionally, the publication dates back to 2003. Therefore the authors should seek more up-to-date data.

Different types of abbreviations are explained in the Abstract and in the Introduction.

In Abstract section are used many abbreviation which are not explained as well as results are mixed with discussion.

The introduction section is mixed with discussion section and vice versa.

SHBG is explained twice (verse - 127 and 130), while LH and FSH not once.

verse 208 – please used once introduced abbreviation HOMA-IR not HOMA.

verse 240 – “Inflammation in lean women with PCOS …” instead “…in lean omen…”.

verse 243 – explain DHEA-C abbreviation

verse 260 – the dot is missed in “et al”

I suggest the division of studied group according to HOMA-IR value, to analyze how insulin resistance affect inflammation.

The discussion section need to be revised with the greater emphasis on comparing the results obtained in the study with those reported by other authors. The discussion section is not well-developed.

In my view, the entire manuscript should undergo a thorough revision, encompassing both editorial and language aspects.

The manuscript investigates different inflammatory and anti-inflammatory parameters in the group of women with and without PCOS divided according to body mass index.

In my opinion the investigated inflammatory and anti-inflammatory cytokines should be treated as parameters not markers.

The results should be compared with commonly used inflammatory markers for general diagnosis, such as CRP or procalcitonin to confirm their usefulness for the determination of inflammatory state.

Furthermore, the chronic inflammatory during PCOS should be treated as chronic low-grade inflammation.  

In title anti- not anty – should be written.

verse 16 - please introduce the abbreviation of PCOS

verse 27 – delete space between p and < (p <0.05)

I can not find in paper [2] information about “About 70% of women with PCOS have IR [2]”. Additionally, the publication dates back to 2003. Therefore the authors should seek more up-to-date data.

Different types of abbreviations are explained in the Abstract and in the Introduction.

In Abstract section are used many abbreviation which are not explained as well as results are mixed with discussion.

The introduction section is mixed with discussion section and vice versa.

SHBG is explained twice (verse - 127 and 130), while LH and FSH not once.

verse 208 – please used once introduced abbreviation HOMA-IR not HOMA.

verse 240 – “Inflammation in lean women with PCOS …” instead “…in lean omen…”.

verse 243 – explain DHEA-C abbreviation

verse 260 – the dot is missed in “et al”

I suggest the division of studied group according to HOMA-IR value, to analyze how insulin resistance affect inflammation.

The discussion section need to be revised with the greater emphasis on comparing the results obtained in the study with those reported by other authors. The discussion section is not well-developed.

In my view, the entire manuscript should undergo a thorough revision, encompassing both editorial and language aspects.

Author Response

Dear Reviewer 1,

Thank you for your time in reading our manuscript, and offering us suggestions to improve it. We deeply appreciate your insightful comments and points of discussion. We have made the following clarifications and changes per your recommendations:

  1. In my opinion the investigated inflammatory and anti-inflammatory cytokines should be treated as parameters not markers.

Thank you for your comment. We have replaced markers with parameters in verses 1,19,35,90,91, 104,146, 147,168, 173,174, 212,243,274,277, 283, 336,344,346-348.

2.The results should be compared with commonly used inflammatory markers for general diagnosis, such as CRP or procalcitonin to confirm their usefulness for the determination of inflammatory state.

Thank you for your suggestion. You are absolutely right, it would add value to our study, however due to budget constraints we are unable to add this data to our study at the moment. We have indicated data from previous studies on these markers in the discussion and also compared them with our data.

Changes in the text: verses 307-316.

We did not analyze the more commonly used inflammatory markers such as CRP or procalcitonin. However, we provide indirect evidence of increased CRP through the increased levels of IL-6 and TNF-a which are the main mediators of CRP production in the liver. According to two large meta-analyses by Escobar-Morreale et al. (2010) and Aboeldalyl et al. (2021), women with PCOS have elevated CRP levels regardless of obesity, which is consistent with our data [9,48]. Procalcitonin is a well-studied marker of bacterial infection [49]. Rashad et al. (2013) showed an increase in procalcitonin levels in women with PCOS that was dependent on the presence of overweight/obesity [50]. Thus, results of our study and other researches provide further evidence supporting the hypothesis that chronic inflammation is related to PCOS pathology independent of obesity.

References:

  1. Aboeldalyl, S., James, C., Seyam, E., Ibrahim, E. M., Shawki, H. E. D., Amer, S. The role of chronic inflammation in polycystic ovarian syndrome—a systematic review and meta-analysis. Int. J. Mol. Sci. 2021, 22, 2734.

48.Escobar-Morreale, H. F., Luque-Ramírez, M., & González, F. inflammatory markers in polycystic ovary syndrome: a systematic review and metaanalysis. Fertil. Steril. 2011, 95,1048-1058.

49.Schuetz, P., Muller, B., Christ‐Crain, M., Stolz, D., Tamm, M., Bouadma, L., Briel, M. Procalcitonin to initiate or discontinue antibiotics in acute respiratory tract infections. Evidence‐Based Child Health: A Cochrane Review Journal. 2013, 8, 1297-1371.

  1. Rashad N. M., Amal S., Abdelaziz A. M. Association between inflammatory biomarker serum procalcitonin and obesity in women with polycystic ovary syndrome. J. Reprod. Immunol. 2013, 97, 232-239.

3.Furthermore, the chronic inflammatory during PCOS should be treated as chronic low-grade inflammation.  

Thank you for your point of clarification. We have replaced chronic inflammation with chronic low-grade inflammation in verses 15,32, 50,51, 102,104, 270,274, 278,279,330,342,344.

  1. In title anti- not anty – should be written.

Thank you for your correction. We have changed the title: Inflammatory and Anti-inflammatory parameters in PCOS Patients Depending on Body Mass Index: A Case-Control Study in verse 1.

  1. verse 16 - please introduce the abbreviation of PCOS

Changes in the text: pathogenesis of polycystic ovary syndrome (PCOS)-verse 16.

  1. verse 27 – delete space between p and < (p <0.05)

Changes in the text: both in lean and overweight/obese subgroups (p<0.05).

  1. I can not find in paper [2] information about “About 70% of women with PCOS have IR [2]”. Additionally, the publication dates back to 2003. Therefore the authors should seek more up-to-date data.

You are completely right. Changes in the text: About 75% of women with PCOS have IR [2]-verse 45.

  1. 2. Tosi F., Bonora E., Moghetti P. Insulin resistance in a large cohort of women with polycystic ovary syndrome: a comparison between euglycaemic-hyperinsulinaemic clamp and surrogate indexes. Reprod. 2017, 32, 2515-2521.
  2. Different types of abbreviations are explained in the Abstract and in the Introduction.

Thanks for your comment. Changes in the text: verses 25,26,29,30.

Cytokines: interleukin-1 receptor antagonist (IL -1 Ra), IL- 2, IL- 6, IL- 17 E, IL- 17 A, IL- 18 and macrophage inflammatory protein-1 alpha (MIP-1 alpha) were increased in women with PCOS compared to controls, both in lean and overweight/obese subgroups (p<0.05). Moreover, only lean women with PCOS had higher levels of IL- 1 alpha, IL- 4, IL- 9, IL- 12, IL -13, IL- 15, tumor necrosis factor (TNF) alpha and beta, soluble CD40 and its ligand (SCD40L), Fractalkine (FKN), monocyte-chemotactic protein 3 (MCP-3), and MIP-1 b compared to the control group (p<0.05). IL- 22 was increased in the combined group of women with PCOS (lean and overweight/obese) compared to the control group (p=0.012). Conclusion: Chronic low-grade inflammation is an independent factor affecting the occurrence of PCOS, and does not depend on the presence of excess weight/obesity. For the first time, we obtained data on the increase of such inflammatory parameters as IL-9, MCP-3 and MIP-1a in women with PCOS.

  1. In Abstract section are used many abbreviations which are not explained as well as results are mixed with discussion.

Thanks, we unfolded all abbreviations in the abstract and text.

  1. SHBG is explained twice (verse - 127 and 130), while LH and FSH not once.

Thanks for your comment. We have left the decoding of SHBG in verse 127, LH and FSH are deciphered in the introduction (verse 45 for LH and verse 81 for FSH)

  1. verse 208 – please used once introduced abbreviation HOMA-IR not HOMA.

We have changed in verses:186,218,223,298,339.

  1. verse 240 – “Inflammation in lean women with PCOS …” instead “…in lean omen…”.

Changes in the text: Inflammation in lean women with PCOS could have other pathogenic mechanisms, but most studies of parameters of chronic low-grade inflammation have been conducted in overweight or obese patients with PCOS-verse 273.

  1. verse 243 – explain DHEA-C abbreviation

Changes in the text: Dehydroepiandrosterone Sulfate (DHEA-S)-verse 276.

  1. I suggest the division of studied group according to HOMA-IR value, to analyze how insulin resistance affect inflammation.

Thank you for your suggestion. We have added the part «IR and inflammation in PCOS», where we analyzed inflammatory parameters in subgroups of women with PCOS depending on IR.

Changes in the text: verse 217.

3.5. IR and inflammation in PCOS.

To study the effect of IR on inflammation in PCOS, we divided women with PCOS according to the HOMA-IR index. A value of more than 2.5 confirmed the presence of IR. Of the 44 women with PCOS, IR was detected in 20 participants. The levels of IL-1 RA and IL-6 were higher in the subgroup with IR (p<0.001) (table 4).

Among chemokines a difference was detected only for Eotaxin, which was significantly lower (median 37.4 versus 53.9) in the subgroup with IR (table 5).

  1. The discussion section need to be revised with the greater emphasis on comparing the results obtained in the study with those reported by other authors. The discussion section is not well-developed.

Thank you for your suggestion to improve the discussion. The discussion section was extended. We have paid more attention to the results of previous studies and compared them with the results we obtained. You can read the modified discussion below.

Changes in the text: verses 307-337.

We did not analyze the more commonly used inflammatory markers such as CRP or procalcitonin. However, we provide indirect evidence of increased CRP through the increased levels of IL-6 and TNF-a which are the main mediators of CRP production in the liver. According to two large meta-analyses by Escobar-Morreale et al. (2010) and Aboeldalyl et al. (2021), women with PCOS have elevated CRP levels regardless of obesity, which is consistent with our data [9,48]. Procalcitonin is a well-studied marker of bacterial infection [49]. Rashad et al. (2013) showed an increase in procalcitonin levels in women with PCOS that was dependent on the presence of overweight/obesity [50]. Thus, results of our study and other researches provide further evidence supporting the hypothesis that chronic inflammation is related to PCOS pathology independent of obesity.

It has been also established that insulin sensitivity is reduced by ~40% in PCOS women, independent of obesity [51]; although obesity further impairs insulin metabolism. TNF-a and IL-6, two main adipose tissue-derived inflammatory cytokines, have been implicated in the induction of insulin resistance [52,53].

In our study the relation between IR and PCOS was detected only for IL-6, IL-1 RA and Eotaxin. IL-6 induces insulin resistance by downregulating insulin receptor substrate-1 (IRS-1) phosphorylation and transcription [54,55] and by upregulating suppressor of cytokine signaling 3 (SOCS3), which inhibits insulin receptor [56]. The involvement of IL-1 in regulation of insulin sensitivity is highlighted by a recent observation that the IL-1 blocking agent improved IR in rheumatoid arthritis patients with comorbid Type 2 diabetes (T2D) [57].

A hypothesis explaining causal relation between chronic inflammation and insulin resistance in PCOS was suggested by Shorakae et al. (2018) who investigated the links between insulin resistance, hyperandrogenism, and chronic low-grade inflammation in PCOS in a cross-sectional study. The authors concluded that PCOS-related hyperandrogenism leads to adipose tissue inflammation with subsequent dysregulated adipokine secretion resulting in insulin resistance [58].

 A plausible reason for the lack of association between IR and the majority of cytokines in our study is that the Homa-IR index is not sensitive enough to detect IR, especially in lean people with PCOS. It is also possible that the effects of other inflammatory parameters altered in PCOS are mediated through other mechanisms unrelated to IR.

References:

  1. Escobar-Morreale, H. F., Luque-Ramírez, M., & González, F. inflammatory markers in polycystic ovary syndrome: a systematic review and metaanalysis. Fertil. Steril. 2011, 95,1048-1058.
  2. Schuetz, P., Muller, B., Christ‐Crain, M., Stolz, D., Tamm, M., Bouadma, L., Briel, M. Procalcitonin to initiate or discontinue antibiotics in acute respiratory tract infections. Evidence‐Based Child Health: A Cochrane Review Journal. 2013, 8, 1297-1371.
  3. Rashad N. M., Amal S., Abdelaziz A. M. Association between inflammatory biomarker serum procalcitonin and obesity in women with polycystic ovary syndrome. J. Reprod. Immunol. 2013, 97, 232-239.
  4. O'Driscoll, J. B., Mamtora, H., Higginson, J., Pollock, A., Kane, J., Anderson, D. C. Prospective Study of the Prevalence of Clear-Cut Endocrine Disorders and Polycystic Ovaries in 350 Patients Presenting with Hirsutism or Androgenic Alopecia. Clin. Endocrinol. 1994, 41, 231–236.
  5. Tilg, H., Moschen, A. R. Inflammatory Mechanisms in the Regulation of Insulin Resistance. Mol. Med. 2008, 14, 222–231.
  6. Guilherme, A., Virbasius, J. V., Puri, V., Czech, M. P. Adipocyte Dysfunctions Linking Obesity to Insulin Resistance and Type 2 Diabetes. Nat. Rev. Mol. Cell Biol. 2008, 9, 367–377.

54.Galic, S., Oakhill, J. S.,Steinberg, G. R. Adipose tissue as an endocrine organ. Cell Endocrinol.2010,316,129–139.

  1. Torres‐Leal, F. L., Fonseca‐Alaniz, M. H., Rogero, M. M., Tirapegui, J. The role of inflamed adipose tissue in the insulin resistance. Cell Biochem. Funct. 2010, 28, 623–631.
  2. Senn, J. J., Klover, P. J., Nowak, I. A., Zimmers, T. A., Koniaris, L. G., Furlanetto, R. W., Mooney, R. A. Suppressor of Cytokine Signaling-3 (SOCS-3), a Potential Mediator of Interleukin-6-Dependent Insulin Resistance in Hepatocytes. J. Biol. Chem. 2003, 278, 13740–13746.

57.Ruscitti, P., Ursini, F., Cipriani, P., Greco, M., Alvaro, S., Vasiliki, L., Giacomelli, R. IL-1 inhibition improves insulin resistance and adipokines in rheumatoid arthritis patients with comorbid type 2 diabetes: An observational study. Medicine. 2019,98.

  1. Shorakae, S., Ranasinha, S., Abell, S., Lambert, G., Lambert, E., de Courten, B., Teede, H. Inter-Related Effects of Insulin Resistance, Hyperandrogenism, Sympathetic Dysfunction and Chronic Inflammation in PCOS. Clin. Endocrinol. 2018, 89, 628-633.

Reviewer 2 Report

This study investigates the changes in inflammatory and anti-inflammatory markers in patients with PCOS. For the study, a variety of inflammatory and anti-inflammatory cytokines were assessed in patients with and without PCOS, matched based on age and BMI. The results reported in the manuscript indicate that IL-1Ra, IL-2, IL-6, IL-17E, IL-17A, IL-18 and MIP-1alpha were elevated in women with PCOS regardless of their BMI as compared to matched controls. In lean women with PCOS, only IL-1 alpha, IL-4, IL-9, IL-12, IL-13 , IL-15, TNFalpha, SCD40, MCP-3, and MIP-1b were elevated. The conclusion of the authors is that chronic inflammation is an independent factor affecting the occurrence of PCOS and does not depend on the presence of abnormal BMI. 

The study appears to be properly conducted and the conclusions are supported by the data reported here. No major criticisms were noted at this time. 

A few typos (starting from the title) need to be amended

A few typos need to be amended, starting with the title (Anti- and not anty-)

Author Response

Dear Reviewer 2,

Thank you for your time in reading our manuscript, and offering us suggestions to improve it. We deeply appreciate your insightful comments and points of discussion. We have made the following clarifications and changes per your recommendations:

A few typos need to be amended, starting with the title (Anti- and not anty-).

Thank you for your suggestion. We have changed the title: Inflammatory and Anti-inflammatory parameters in PCOS Patients Depending on Body Mass Index: A Case-Control Study.

Reviewer 3 Report

Comments about the manuscript:

“Inflammatory and Anty-inflammatory Markers in PCOS Patients Depending on Body Mass Index: A Case-Control Study.”

Chronic inflammation is strongly implicated in polycystic ovary syndrome, but the cytokines that influence this syndrome are still not known, nor their link with excess weight. This manuscript concerns research into markers (32 cytokines measured in plasma) of chronic inflammation in 44 women with polycystic ovary syndrome compared to 45 healthy women. The study involves both normal weight and overweight people.

This work provides useful results and should be able to be published after some minor corrections.

Page 4, lines 152-153. “according to the manufacturer's instructions” does not seem sufficient for a scientific article: briefly describe the method used.

Page 4, line 164. “Clinical, biochemical and hormonal parameters”: in this part, it would be useful to give several results in the text (for example between brackets).

Page 5, legend of table 1: “MS menstrual cycle”: replace “MS” with “MC” (like in the table).

Page 10, line 253 : write “Peng et al. (2016)” instead of “Peng Z et al. (2016)”.

Page 10, line 252 “Neher et al. (2016)” : is it “Neher” or “Nehir” like in reference list. PLease, check.

Page 10, line 264: write “Talaat et al. (2016)” instead of “Talaat R. M. et al. (2016)”.

Page 10, line 269 : write “Mehrabani et. al. (2012)” instead of “Mehrabani H. H. et. al. (2012)”.

Author Response

Dear Reviewer 3,

Thank you for your time in reading our manuscript, and offering us suggestions to improve it. We deeply appreciate your insightful comments and points of discussion. We have made the following clarifications and changes per your recommendations:

  1. Page 4, lines 152-153. “according to the manufacturer's instructions” does not seem sufficient for a scientific article: briefly describe the method used.

Thank you for your comment. We have described the method.

Changes in the text: Luminex® uses proprietary techniques to internally color-code microspheres with two fluorescent dyes. Using the concentration of these dyes, sets of beads coated with capture antibody were created. After capturing the analyzed sample by the bead, a biotinylated detection antibody was injected. The reaction mixture was then incubated with the streptavidin-PE conjugate to complete the reaction on the surface of each microsphere. Each individual microsphere was identified and the result of its bioassay is quantified based on fluorescent reporter signals.

  1. Page 4, line 164. “Clinical, biochemical and hormonal parameters”: in this part, it would be useful to give several results in the text (for example between brackets).

Thank you for your suggestion. We have added to the description in brackets.

Changes in the text: data of clinical (age, BMI), biochemical (FPG, total cholesterol), hormonal parameters (insulin, sex hormones

  1. Page 5, legend of table 1: “MS menstrual cycle”: replace “MS” with “MC” (like in the table).

We have replaced.

  1. Page 10, line 253 : write “Peng et al. (2016)” instead of “Peng Z et al. (2016)”.

We have rewritten without Z.

  1. Page 10, line 252 “Neher et al. (2016)” : is it “Neher” or “Nehir” like in reference list. Please, check.

We have changed as in reference.

  1. Page 10, line 264: write “Talaat et al. (2016)” instead of “Talaat R. M. et al. (2016)”.

We have rewritten without R.M.

  1. Page 10, line 269 : write “Mehrabani et. al. (2012)” instead of “Mehrabani H. H. et. al. (2012)”.

We have rewritten without H.H.

Round 2

Reviewer 1 Report

In current version I accept the paper